# Negative Regulation of FGFR (Fibroblast Growth Factor Receptor) Signaling

**DOI:** 10.3390/cells10061342

**Published:** 2021-05-28

**Authors:** Patrycja Szybowska, Michal Kostas, Jørgen Wesche, Ellen Margrethe Haugsten, Antoni Wiedlocha

**Affiliations:** 1Department of Tumor Biology, Institute for Cancer Research, The Norwegian Radium Hospital, Oslo University Hospital, Montebello, 0379 Oslo, Norway; Patrycja.Szybowska@rr-research.no (P.S.); Michal.Janusz.Kostas@rr-research.no (M.K.); Jorgen.Wesche@rr-research.no (J.W.); 2Centre for Cancer Cell Reprogramming, Institute of Clinical Medicine, Faculty of Medicine, University of Oslo, Montebello, 0379 Oslo, Norway; 3Department of Molecular Cell Biology, Institute for Cancer Research, The Norwegian Radium Hospital, Oslo University Hospital, Montebello, 0379 Oslo, Norway; 4Military Institute of Hygiene and Epidemiology, 01-163 Warsaw, Poland

**Keywords:** FGFR, signaling, endocytosis, negative regulation, phosphatases

## Abstract

FGFR (fibroblast growth factor receptor) signaling controls fundamental processes in embryonic, fetal and adult human life. The magnitude, duration, and location of FGFR signaling must be strictly controlled in order to induce the correct biological response. Uncontrolled receptor signaling has been shown to lead to a variety of diseases, such as skeletal disorders and cancer. Here we review the numerous cellular mechanisms that regulate and turn off FGFR signaling, once the receptor is activated. These mechanisms include endocytosis and endocytic sorting, phosphatase activity, negative regulatory proteins and negative feedback phosphorylation events. The mechanisms act together simultaneously or sequentially, controlling the same or different steps in FGFR signaling. Although more work is needed to fully understand the regulation of FGFR signaling, it is clear that the cells in our body have evolved an extensive repertoire of mechanisms that together keep FGFR signaling tightly controlled and prevent excess FGFR signaling.

## 1. Introduction

The pleiotropic biological actions of FGFs (fibroblast growth factors) are exerted through binding to and activation of four conserved, transmembrane cell-surface FGFRs (fibroblast growth factor receptors) with tyrosine kinase activity, named FGFR1-4. In humans, 22 structurally related FGFs have been identified based on sequence homology [1]. All FGFs are recognized by a highly conserved core of around 140 amino acids that exhibits a beta-trefoil structure [2,3]. The beta-trefoil protein fold belongs to the oldest evolutionary protein folds [4]. FGFs probably originated from a beta-trefoil FGF-like domain present in a choanoflagellate metazoan ancestor [5,6]. Eighteen of these FGFs function as high affinity ligands for the four FGFRs and multiple splice variants of three of them (FGFR1-3). FGFRs consist of two or three extracellular immunoglobulin (Ig)-like domains, a transmembrane domain and an intracellular split tyrosine kinase domain. Of particular importance is the alternative splicing of FGFR1-3 at the third Ig-like domain giving rise to FGFR1-3 b or c forms with altered affinity for different FGFs and alternative expression patterns [7]. While the FGFRb forms are generally expressed in epithelial cells, the FGFRc forms are generally found in mesenchymal cells [8].

The FGFs act as tissue growth factors (canonical FGFs) or metabolic hormones (endocrine FGFs) [9] and form together with the different FGFRs a regulatory system that is operative in vertebrates as well as in invertebrates, controlling fundamental processes in embryonic, fetal and adult life [1,10]. FGF/FGFR signaling is an ancient metazoan cell communication mechanism predating the cnidarian-bilaterian divergence to even before the Cambrian explosion events, over half a billion years ago and is expressed by all extant taxa surveyed up to now [5,11]. The FGF/FGFR families have evolved through gene amplification and differentiation as a regulatory signaling system crucially important for a sophisticated organization of tissues in multi-cellular organisms [1,12,13]. It appears that FGF-induced signaling is highly adaptive and promotes/facilitates the possibilities of great variety in life forms [1,14].

In vertebrates, FGFs/FGFRs form one of the largest protein signaling families. Already from this point of view, it is not surprising that aberrations in the FGF/FGFR axis signaling are often found in developmental and metabolic disorders as well as in highly malignant diseases like breast, lung and prostate cancers [15,16,17]. Deregulation in the control of FGFR signaling also leads to skeletal disorders. Mutations in FGFR3 cause achondroplasia, a frequent form of dwarfism [18]. Additionally, mutations in FGFR1-3 can result in development of chondrodysplasia, craniosynostosis and skeletal overgrowth syndromes [19]. Knowledge of the regulatory mechanisms that govern FGF signaling is important in order to understand many aspects of FGF/FGFR biology and disease.

The binding of FGF ligands to their specific receptors results in receptor dimerization and conformational changes in the cytoplasmic part of the receptor. The conformational changes lead to trans-autophosphorylation of the tyrosine kinase domains of the receptors and subsequently the induction of several downstream signaling pathways. There are four major signaling pathways activated by FGF/FGFRs; the Ras-MAPK (mitogen-activated protein kinase) pathway, the PI3K (phosphatidylinositol-4,5-bisphosphate 3-kinase)-AKT pathway, the PLCγ (phospholipase Cγ)/PKC (protein kinase C) pathway and the STAT (signal transducer and activator of transcription) pathway [9] (Figure 1). There are several excellent reviews on FGFR signaling and therefore, this will not be discussed in further detail here [9,16,20]. We will focus here on how FGFR signaling is inhibited and turned off, once the receptor is activated.

Accurate signaling from receptor tyrosine kinases can be maintained through tight regulation by several mechanisms [21]. However, in contrast to the well-studied mechanism of FGFR activation and signaling, the mechanisms ensuring receptor deactivation are less understood. Downregulation of FGFR signaling has been reported to occur through mechanisms involving phosphatases, negative regulator proteins, and negative feedback phosphorylations (Figure 1). However, the main mechanism that regulates the duration and strength of receptor tyrosine signaling is endocytosis (Figure 1 and Figure 2).

## 2. Endocytosis

Endocytosis is a cellular process involving membrane invagination and uptake of different cargos such as receptor-ligand complexes, antigens, viruses and protein toxins into intracellular vesicles [22]. Vesicles and tubular structures with endocytic cargo are fused with early/sorting endosomes from which the cargo is sorted to different intracellular destinations (Figure 2). Endocytosed receptors might be transported into intraluminal vesicles in multivesicular bodies/late endosomes/lysosomes for degradation resulting in attenuation of signaling while some receptors are recycled back to the plasma membrane, leading to sustained signaling [23]. In the case of FGFR, endocytosis regulates signaling in several ways and depending on the receptor type, ligand type and possibly also cell type, the receptors are sorted differently resulting in altered signaling. We will here go through the mechanisms of FGFR endocytosis and how endocytosis influences FGFR signaling. 

The first step in receptor-mediated endocytosis is the recruitment of active receptor-ligand complexes into growing invaginations at the plasma membrane that eventually buds off to form intracellular vesicles. Receptor internalization is often divided into clathrin-mediated endocytosis (CME) and clathrin-independent endocytosis (CIE). CIE pathways include caveolae-mediated, FEME (fast endophilin-mediated endocytosis) and CLIC/GEEC (clathrin-independent carriers/GPI-enriched early endosomal compartments) [24]. CME is the main route for many cell surface receptors and their ligands [25]. CME is characterized by the formation of clathrin coated pits at the plasma membrane which pinch off from the cell surface by the large GTPase dynamin. In the case of FGFRs, FGFR1, FGFR2 and FGFR4 are mainly internalized via CME, while FGFR3 internalization involves both CME and CIE processes [26,27,28,29,30,31,32] (Figure 2). In the case of FGFR3, it is not clear which mechanisms are involved in CIE, but dynamin seems not to be required [27,32]. It is worth mentioning that FGFRs have been observed in caveolae but it is not clear if they are internalized via caveolin-mediated endocytosis [33,34]. Additionally, FGF1 stimulation seems to induce the formation of an endophilin positive structure at the plasma membrane, indicating that FEME pathway might be involved in FGF1 uptake in some cells [35]. It is, however, not clear which FGFRs are expressed in these cells.

In the case of another RTK (receptor tyrosine kinase), namely EGFR (epidermal growth factor receptor), the receptor is taken up via both CME and CIE depending on low (<2 ng/mL) or high (>2 ng/mL) ligand concentration, respectively [36]. At even higher ligand concentrations (>50 ng/mL), EGFR is internalized via FEME [24,37]. It is not clear how this applies to other RTKs and FGFRs. In the case of FGFR1 (CME endocytosis) and FGFR3 (partial CME and CIE), ligand concentrations did not alter their dependency on clathrin for endocytosis [27]. 

It is uncertain what might be the signal for recruitment of FGFRs to invaginations at the plasma membrane leading to their endocytosis. Most receptors internalized via CME are recruited to the clathrin-coated membrane invaginations via tyrosine-based sequences such as YXXФ (where Ф can be any hydrophobic residue) or [FY]XNPX[FY] motives [38]. These sorting signals bind directly to the µ2 subunit of AP2, a major adaptor protein in CME, or other adaptor proteins involved in CME that contain phosphotyrosine binding (PTB) domains. Although some putative signal sequences of both types are present in FGFR1-4 (our unpublished data), none of these has yet been experimentally demonstrated to function as endocytic signals. However, a few factors seem to be crucial for FGFR endocytosis. Inhibition of receptor kinase activity results in reduced internalization, indicating that ligand binding, dimerization and trans-autophosphorylation seem to be required for proper endocytosis. An inactive, kinase dead FGFR1 (K514R) as well as treatment with FGFR inhibitors drastically reduced the rate of endocytosis [26,31,33,39,40]. In addition, the mutation of tyrosine 766 in FGFR1, which is the binding site for PLC-γ in activated FGFR1, also reduced receptor internalization [40]. However, in the case of FGFR2, tyrosine 769 (corresponding to tyrosine 766 in FGFR1) is not required for endocytosis [41]. It has also been demonstrated that phosphorylation of a specific serine (S789) in FGFR1 by RSK2 (ribosomal s6 kinase 2) is important for internalization [42] and functions as a negative feedback loop since RSK2 is activated by the Ras-MAPK pathway downstream of FGFR1 (see chapter below). 

Recruitment of E3 ubiquitin ligases such as NEDD4-1 (neural precursor cell expressed developmentally down-regulated protein 4) and c-CBL (Casitas B-lineage lymphoma) either directly to the receptor or via FRS2 (FGFR substrate 2)/Grb2 (growth factor receptor-bound 2), respectively, and subsequent ubiquitination of the receptor has also been implicated in FGFR endocytosis [43,44,45]. However, a lysine mutant of FGFR1 lacking most of the potential ubiquitination sites, was endocytosed to a similar degree as wild type FGFR1 indicating that ubiquitination is not crucial for internalization [26]. Several other proteins have been implicated in FGFR endocytosis. For instance, proper kinetics of FGFR2 internalization seems to depend on the recruitment and activation of Src and Eps8 (epidermal growth factor receptor kinase substrate 8) but not Eps15 (epidermal growth factor receptor kinase substrate 15) [31,46]. In the case of endocytosis of FGFR1, interaction between FGFR1 and Esyt2 (extended synaptotagmin-2) appears to be required for proper CME [47]. Another protein implicated in FGFR endocytosis is the tumor suppressor, VHL (von Hippel-Lindau protein). Loss of VHL led to accumulation of FGFR1 at the cell surface and enhanced signaling due to impaired internalization [48,49]. Additionally cell adhesion molecules such as N-cadherin and E-cadherin seem to play a role in FGFR endocytosis. Binding of FGFR1 to N-cadherin or overexpression of E-cadherin, can delay FGFR1 endocytosis [50,51]. Syndecan-4, a heparan sulfate proteoglycan at the cell surface, is involved in macropinocytosis of FGFR1 [52]. Although recruitment of several proteins to active FGFRs seems to play a role in FGFR endocytosis, it has also been suggested that FGFR1 dimerization rather than activation is required for CME [53]. To this end, it is not completely clear how the different findings regarding recruitment of FGFRs to the endocytic machinery, choice of machinery and proteins involved add together. More research, preferentially under similar experimental settings, is needed in order to fully understand these processes and to elucidate receptor or ligand determined preferences.

Once internalized, the number of receptors available for binding of ligands at the cell surface is reduced and the cells are less responsive to ligand. The receptor can continue to signal from endosomes and it has been suggested that different intracellular locations might give rise to different signaling simply because the subsets of downstream signaling molecules might vary from compartment to compartment [54]. For example, in cells depleted for Rab11, a master regulator of recycling, FGFR4 accumulates intracellularly leading to sustained PLC-γ signaling, but reduced AKT signaling [55]. It seems that recycling is needed for FGFR4 to interact with the AKT signaling machinery to maintain AKT signaling. On the other hand, PLC-γ that binds directly to the receptor is localized with FGFR4 and signaling is sustained even if the receptor is trapped in endosomes [55]. When endocytosis of FGFR1 is reduced, due to depletion of cells for clathrin, not only prolonged MAPK signaling was observed but also a delay in activation [27]. This indicates that endocytosis of FGFR1 is needed for full activation of MAPK signaling as well as for downregulation of MAPK signaling [27]. Delayed endocytosis of FGFR1 due to binding to N-cadherin, also led to sustained MAPK signaling [50]. On the other hand, sustained MAPK signaling was observed upon FGFR1 macropinocytosis by removal of syndecan-4 from cells [52]. Additionally, trapping of FGFR1 at the cell surface upon E-cadherin overexpression, reduced MAPK signaling [51,52]. As is evident from these studies, it is not entirely clear how endocytosis influences FGFR-induced MAPK signaling. The effect of endocytosis and different intracellular localization of FGFRs on signaling might be cell type dependent or even dependent on the experimental conditions. It is however clear, that FGFRs can signal not only from the plasma membrane but also from endosomes and that their signaling properties might vary with their localization (for instance plasma membrane versus endosomal location). It is also worth mentioning that FGFs and FGFRs can traffic to the nucleus (reviewed in [56,57]). Although their function in the nucleus is not clear, nuclear localization seems to be required for promigratory effect of FGF stimulation in breast and pancreatic cancer cells, as well as HIF (hypoxia-inducible factor)-mediated hypoxic responses in prostate cancer cells [58,59,60].

Following endocytosis, receptors are transported to the lysosomes for degradation resulting in termination of signaling or for recycling back to the cell surface allowing additional rounds of signaling [23,61] (Figure 2). The newly formed endocytic carrier vesicles originating from the plasma membrane undergo fusion events to form early endosomes also called sorting endosomes and it is here, at the early/sorting endosome, the fate of the internalized receptors are decided. The receptors at the early/sorting endosomes are either sorted for lysosomal degradation or they are retrieved into tubule-vesicular transport carriers for recycling back to the plasma membrane [23,61]. Recycling back to the plasma membrane can occur via the endosomal recycling compartment in a Rab11 dependent manner (slow recycling) or directly from early/sorting endosomes back to the plasma membrane in a Rab4 dependent manner (fast recycling) [23]. Receptors destined for degradation are sorted into intraluminal vesicles originating at the early/sorting endosomal membrane. This occurs as the early endosomes mature into a late endosome [23]. Late endosomes are characterized by the appearance of multiple intraluminal vesicles and are also referred to as multivesicular bodies. Late endosomes usually fuse to lysosomes resulting in degradation of their content [23]. 

The pathway that FGFRs follow after internalization depends on both the receptor-type but also the bound ligand [62] (Figure 2). Upon stimulation with FGF1, FGFR1-3 are mainly sorted to lysosomes, but with lower efficiency in the case of FGFR2 and FGFR3 while FGFR4 is mainly recycled leading to sustained signaling [62]. Stimulation of FGFR2b with FGF7 also resulted in lysosomal sorting and degradation while stimulation with FGF10 led to recycling and increased signaling [29,63]. Moreover, FGFR1 bound to NCAM (neural cell adhesion molecule), an unconventional ligand for FGFR1, led to FGFR1 recycling [64]. Similarly, NEGR1 (neuronal growth regulator 1) seems to be important for FGFR2 recycling [65]. The recycling pathway might also involve transport via the TGN (trans Golgi network) as internalized FGF1/FGFR4 complexes were found to localize partially with TGN structures [28]. FGF/FGFRs can also be transported to the nucleus [59,66,67].

While the retrieval of receptors for recycling is poorly understood, sorting of receptors into intraluminal vesicles and the degradative pathway is well characterized. The key step for sorting of a receptor into intraluminal vesicles and subsequent degradation, is the attachment of ubiquitin to lysine residues in the intracellular part of the receptor by E3 ubiquitin ligases such as CBL and NEDD4 [23,68]. The ESCRT (endosomal sorting complex required for transport) complexes are recruited to the endosomal membrane and concentrate ubiquitinated proteins in degradative subdomains that eventually bud off to form intraluminal vesicles (Figure 2). The ESCRT complexes, ESCRT-0, -I, -II and -III operate in a sequential manner [69]. First, ESCRT-0 consisting of HRS (hepatocyte growth factor-regulated tyrosine kinase substrate) and STAM 1 (signal transducing adaptor molecule 1), each binding with low affinity to ubiquitin, cluster the ubiquitinated cargo. This leads to recruitment of ESCRT-I and ESCRT-II. Accumulation of ESCRT-II at the degradative subdomains leads to recruitment of ESCRT-III. ESCRT-III together with VPS4 (vacuolar protein sorting-associated protein 4) generate inwardly budding of the membrane followed by scission to generate an intraluminal receptor-containing vesicle. Before scission, the receptors are deubiquitinated and the ESCRT-0-II is dissociated [23,69,70].

To understand the different sorting of FGFR1 and FGFR4, a comparison of their intracellular sequences was performed and revealed that FGFR1 contains a higher number of lysines that are potential ubiquitination sites, than FGFR4 [62]. Indeed, it was demonstrated that FGFR1 was more efficiently ubiquitinated than FGFR4. Moreover, removing potential ubiquitination sites in FGFR1 by substituting lysines for arginines, led to reduced ubiquitination of FGFR1 and forced FGFR1 to recycle [26]. It has also been demonstrated that the binding of N-cadherin to FGFR1 reduces ubiquitination of FGFR1 resulting in increased FGFR1 stability [71]. The sorting of FGFR2b to degradation upon binding to FGF7 seems to depend on HRS (a component of ESCRT-0), TSG101 (tumor susceptibility gene 101 protein) (a component of ESCRT-I), and ubiquitination [29,72]. FGF7 induced FGFR2b ubiquitination to a greater extent than FGF10 possibly explaining why FGF7 induces lysosomal routing while FGF10 induces receptor recycling. The reduced ubiquitination of FGFR2b upon FGF10 stimulation might result from less efficient phosphorylation of FRS2 leading to reduced recruitment of the ubiquitin ligase c-CBL (discussed below) [29]. It was also reported that stimulation with FGF10 in contrast to FGF7 led to the phosphorylation of Y734 in FGFR2b and subsequent recruitment of PI3K and SH3BP4 (SH3-binding protein 4) [63]. Both phosphorylation of Y734 and the presence of SH3BP4 were required for FGF10 induced FGFR2b recycling. Interestingly, FGFR3 harboring mutations associated with skeletal disorders was found to escape lysosomal targeting due to reduced levels of ubiquitination [73]. On the other hand, decreased degradation but excessive ubiquitination of FGFR3 harboring these mutations associated with skeletal disorders, has also been reported [74,75] indicating that there might exist alternative mechanisms which independently of ubiquitination levels allow the receptors to escape into the recycling pathway. Although other factors might be involved, it seems that different levels of FGFR ubiquitination, as for many RTKs, might be the main mechanism deciding their fate at the early/sorting endosome.

It is not completely clear which ubiquitin ligases are involved in FGFR ubiquitination but many reports have implicated c-CBL [29,43,73,76,77]. It has been shown that Grb2 interacts with CBL and is recruited to phosphorylated FRS2α in an FGF-dependent manner [29,43]. In this way, CBL recruitment functions as a negative feedback loop in FGFR signaling. Interestingly, in the case of FGFR2b, FGF7 stimulation (leading to receptor degradation) also resulted in a more efficient recruitment of c-CBL than FGF10 stimulation (leading to receptor recycling) [29]. Ubiquitination of FGFR has been shown to increase with overexpression of CBL [43]. It is not clear which receptor was examined in these experiments but similar effects have been observed in the case of FGFR3 [73]. Likewise, overexpression of a dominant negative form of CBL reduced FGFR2 ubiquitination [78]. Additionally, lysosomal transport of FGF8 was delayed upon interfering with the function of CBL [79]. On the other hand, it has been reported that overexpression of wild-type c-CBL or a dominant negative c-CBL variant did not significantly alter FGFR3 ubiquitination [74]. Interestingly, FGFR3 has also been suggested to be stabilized by binding to Hsp90 (heat shock protein 90) [80,81] and it was shown that FGFR3 was ubiquitinated by CHIP (carboxyl terminus of HSP70-interacting protein) ubiquitin ligase probably leading to degradation through a proteasomal pathway rather than the endolysosomal pathway [80]. In addition, NEDD4-1 has been shown to directly bind to FGFR1 leading to ligand induced ubiquitination [44,45]. Under these experimental conditions, the knockdown of CBL did not alter the stability of active FGFR1 [44]. Taken together, several ubiquitin ligases might be involved in FGFR ubiquitination leading to receptor degradation.

Clearly, endocytosis and intracellular sorting influence receptor signaling in many ways. Receptor trafficking is not only altering the duration of signaling, but also the signaling pathways activated might vary with subcellular localization. This might give rise to a completely different signaling output. For instance, altered intracellular sorting of FGFR2b upon stimulation with FGF10 (recycling) reduced signaling duration and led to decreased breast cancer cell migration and inhibition of epithelial branching compared to stimulation with FGF7 (degradation) [63]. Similarly, in contrast to FGF2-induced degradation of FGFR1, NCAM promoted recycling of the receptor, leading to a recycling-dependent increase in cell migration [64]. Expression of ubiquitination-deficient FGFR1 mutants in adult dorsal root ganglia cells led to enhanced recycling and increased axon elongation without stimulating axon branching compared to FGFR1 wild-type [82]. By contrast, the inhibition of FGFR1 endocytosis reduced axon elongation and enhanced axonal branching [83]. Trapping FGFR1 at the cell surface as well as forcing FGFR1 to recycle (ubiquitination-deficient FGFR1 mutants), led to increased signaling but had opposite effects on axon growth. In this case, receptor localization/routing rather than the duration decided the signaling output. 

Most studies on FGFR endocytosis and signaling are performed in cell lines. The understanding of the mechanisms and consequences of FGFR endocytosis in vivo is limited. It has been shown that expression of a mutant form of FGFR1, which is unable to bind to NEDD-4 leading to reduced endocytosis and sustained signaling, results in disrupted anterior neuronal patterning (head development) in zebrafish [44]. Another example is the role of endocytosis in FGF8 morphogen gradient formation [79,84]. FGF8 acts as a diffusible morphogen during vertebrate development and has key roles in a variety of developmental processes, including limb and brain development. In the nascent neuroectoderm of living zebrafish embryos, the spread of FGF8 through the tissue is controlled by endocytosis and subsequent degradation in lysosomes [84]. Upon inhibition of endocytosis, FGF8 accumulated extracellularly and spread over a greater distance in tissue while enhanced FGF8 internalization reduced the signaling range. The effects seemed to be dependent on CBL-mediated lysosomal transport of FGF8 [79]. Yet another example showing the importance of FGFR regulation by endocytosis in vivo is the regulation of FGFR2 by the cell adhesion protein, NEGR1. The knockout of NEGR1 or FGFR2 affected neuronal migration and spine density during mouse cortical development and resulted in impaired core behaviors related to autism spectrum disorders [65]. In cell lines, removal of NEGR1 led to increased degradation of FGFR2 and less signaling. In mice, overexpression of FGFR2 could rescue the effect of NEGR1 knockout indicating that the effect on mouse cortical development was due to decreased FGFR2 protein levels and signaling [65]. Taken together, endocytosis and intracellular transport regulate FGFR signaling giving rise to alternative biological outputs.

## 3. Phosphatases

Protein tyrosine phosphatases (PTPs) remove the phosphate group from tyrosine residues of phosphorylated proteins [85] and as such are primary suspects as negative regulators of FGFR activity. However, to date only PTPRG (protein tyrosine receptor-type G) has been found to inhibit the activity of FGFR1 by direct dephosphorylation of activated FGFR1 [86]. The phosphatase also downregulates FGFR2-4 phosphorylation. It was estimated that PTPRG accounts for ~80% of all phosphatases activity in the early stage of FGFR1 activation [86]. Cancer cells depleted of PTPRG display increased FGFR activity and are hypersensitive to stimulation by FGF1. Moreover, PTPRG depletion elevated cell growth and negatively affected the efficacy of FGFR kinase inhibitors [86]. Due to the efficient dephosphorylation of FGFRs by PTPRG (~80% for FGFR1), other PTPs involved in direct dephosphorylation and negative regulation of FGFRs may be difficult to find due to the functional redundancy and a small share in the process. However, FGFR3 is subject to context-dependent regulation by the phosphatases, PTPN1 and PTPN2 (protein tyrosine phosphatase non-receptor type 1 and 2), and loss of either PTP resulted in ligand-independent activation of FGFR3 [87]. Interestingly, while PTPRG seems to dephosphorylate FGFRs mainly at the cell surface, PTPN1 is localized to the ER and prevents FGFR3 phosphorylation during ER-Golgi processing [86,87].

Phosphatases can also play a role in the positive regulation of FGFR signaling. The best example is the phosphatase SHP2 (Src homology region 2 domain-containing phosphatase 2, also known as PTPN11 (protein tyrosine phosphatase non-receptor type 11)), which binds to phosphorylated FRS2 upon FGF stimulation, but enhances FGFR signaling [88,89]. Once recruited to FRS2, SHP2 is tyrosine phosphorylated and recruits Grb2 leading to the initiation of downstream signaling pathways [89]. Recruited SHP2 can dephosphorylate and inactivate SPRY (Sprouty) (discussed below) thereby causing dissociation of Grb2 from SPRY and suppressing negative impact of SPRY on signaling activation [88]. Another interesting example is the inositol phosphatase, SHIP2 (phosphatidylinositol-3,4,5-trisphosphate 5-phosphatase). SHIP2 has been shown to bind activated FGFRs and nucleate recruitment of Src-family kinases, leading to prolonged activation of FGFRs [90]. Yet, this function of SHIP2 is independent of its phosphatase activity, and rather depends on scaffolding properties by bringing in Src-family kinases as positive regulators. Phosphatases targeting downstream effectors of activated FGFRs are additional players in the regulation of FGFR activity. A known example is MKP3 (MAPK phosphatase 3 also known as DUSP6 (dual-specificity phosphatase 6)), which is an ERK (extracellular signal-regulated kinase)-specific phosphatase encoded by the gene *Dusp6*. MKP3 acts as a key modulator and controls MAPK deactivation by dephosphorylating ERK1 and ERK2 on phosphotyrosine and phosphothreonine residues [91]. Several studies show that *Dusp6* transcription is activated by FGF signaling and suggest a negative feedback role for MKP3 in FGF signaling [92,93,94]. FGFR negative feedback regulation is driven by direct binding of the ERK1/2-responsive transcription factor ETS2 to the *Dusp6* promoter [95]. Downregulation of FGF-induced MAPK signaling by MKP3 seems to play critical roles in regulating developmental outcomes in vertebrates. For example, loss of MKP3 in mice embryos resulted in partially penetrant postnatal lethality, skeletal malformations, and hearing loss, phenotypes that are characteristic of activating mutations in FGFRs [91]. Overexpression of MKP3 in chick embryos reduced levels of activated MAPK in the neural plate and retarded limb bud outgrowth [93]. FGF8-induced MKP3 expression seems to be important during chick, mouse and zebrafish limb/fin development [94]. 

## 4. Negative Regulatory Proteins

A class of well-studied molecules that regulate FGFR signaling are the SPRY proteins (Figure 1). Four members of the SPRY (SPRY1-4) family were discovered as ligand-inducible antagonists of RTK signaling [21]. RTK signaling can influence SPRY proteins by increasing SPRY expression, regulating recruitment to the plasma membrane and modulating SPRY activity by transient tyrosine phosphorylation [96]. ERK1/2 activity has an impact on the expression of SPRY proteins as in several studies SPRY expression was abolished by MEK (mitogen-activated protein kinase kinase) inhibition [97]. SPRY proteins regulate FGFR signaling at various levels and in several ways. First of all, upon FGF stimulation, SPRY1 and SPRY2 translocate to the plasma membrane where they are phosphorylated on a conserved N-terminal tyrosine residue (Y53 or Y55, respectively) and interact with Grb2 [98]. The binding of SPRY1/2 to Grb2 prevents Grb2 from binding to either FRS2 or SHP2 and consequently inhibits MAPK signaling [98]. In addition, SPRY2 has also been shown to bind and inhibit the activity of Raf, a serine/threonine kinase that functions downstream of Ras [99]. There are probably at least two distinct pools of SPRY2, one that binds PP2A (protein phosphatase 2A) and another that binds c-CBL [100]. PP2A binding is required for dephosphorylation of SPRY2, while direct interaction with c-CBL possibly directs SPRY2 for degradation via ubiquitination [100]. Overexpression of SPRY1 and 2 resulted in decreased ubiquitination and increased stability of FGFR2 and FGFR3, respectively [101,102]. The authors suggested that overexpression of SPRY proteins sequesters CBL away from FGFR:FRS2:Grb2 complexes and thus prevents FGFR ubiquitination. In PC12 cells, the activity and binding of CK1 (casein kinase 1) to SPRY2 was necessary for the inhibitory function of SPRY2 in FGFR signaling by promoting binding of SPRY2 to Grb2 [103]. SPRY2 has been shown to act as a key regulator of FGFR signaling in fetal lung development as SPRY*2* knockout rats had severe defects in lung morphogenesis [104]. Contrary, TESK1 (testicular protein kinase 1) and DYRK1A (dual-specificity tyrosine phosphorylation-regulated kinase 1A) can promote FGFR signaling by reversing the inhibitory effects of SPRY2 in both in vivo and in vitro models [105,106]. SPRY4, another member of the SPRY protein family, can also significantly abolish FGF2-induced ERK activation by sequestering Sos1 (Son of sevenless homolog 1) [107]. Interestingly, hetero-oligomers that formed between SPRY1 and SPRY4 more effectively suppressed the ERK activation by inhibiting the association of the Grb2-Sos1 complex with FRS2 [107]. Taken together, SPRY proteins repress FGFR signaling through multiple mechanisms, which likely depend on the cellular context, growth factor- and receptor-type and/or the experimental conditions.

SPRY share several features with the SPRED (Sprouty related with EVH1 (Ena/VASP homology 1)) protein family composed of SPRED-1, SPRED-2, and SPRED-3. Both protein families have a homologous cysteine-rich C terminus and function as inhibitors of the Ras-MAPK pathway downstream of various stimuli like growth factors and cytokines [108,109]. As for SPRY proteins, the mechanisms by which the SPRED proteins negatively regulate Ras signaling are not entirely clear. One way that SPRED proteins suppress Ras signaling is by the recruitment of a the Ras GTPase activating protein, NF-1 (neurofibromin) to the plasma membrane [110]. The recruitment of NF-1 leads to hydrolysis of GTP to GDP in Ras and inactivation of the Ras-MAPK signaling pathway. Other reports suggest that SPRED proteins downregulate Ras signaling by directly preventing Raf phosphorylation and activation [111]. In the case of FGFR signaling, increased ERK phosphorylation upon FGF stimulation was observed in SPRED2-deficient mice [112]. Simultaneously, a dwarf phenotype similar to human achondroplasia, usually caused by activating mutations in FGFR3, was also observed in the SPRED2 knockout mice. Loss of SPRED2 seems to inhibit chondrocyte proliferation due to increased FGF-induced ERK activation, resulting in reduced bone growth [112]. Moreover, SPRED2 seems to play a role in FGFR degradation through direct binding to late endosomal protein NBR1 (neighbor of BRCA1 gene 1 protein) [113]. Similarly to SPRY, SPRED2 interacts with CBL and this interplay controls protein levels of SPRED2 as ubiquitination by CBL targets SPRED2 for degradation [114]. The binding of p85, a subunit of PI3K, to SPRED2 augments the SPRED2-mediated inhibitory effect by increasing Ras binding to SPRED2 and decreasing SPRED2 ubiquitination [115]. 

Another molecule that acts as an antagonist of FGFR signaling is SEF (similar expression to FGF), a conserved inhibitor of the MAPK pathway [116]. However, the precise role of SEF in FGFR signaling is still not clear. SEF has been proposed to act directly in the MAPK signaling pathway as well as at the FGFR itself. In the MAPK pathway, SEF might act at several points. Reports have shown that SEF acts downstream of or at MEK level by inhibiting ERK phosphorylation [117]. On the contrary, it was also reported that SEF can act upstream of MEK possibly at the level of Ras in prostate cancer cells [118]. It has been proposed that SEF directly interacts with Ras at the plasma membrane causing inhibition of Ras activation by FGFR [119]. Results showing that SEF interacts with FGFR across different species and cell types and can inhibit FGFR phosphorylation support the observation that SEF might inhibit components early in the signaling cascade (prior to Ras activation) [119,120]. Direct binding of SEF to FGFR might influence receptor dimerization and consecutive trans-autophosphorylation or can modify receptor-ligand interaction leading to attenuation of signal transduction [120]. The finding that SEF acts on the receptor level or at early steps in FGFR signaling and inhibits various signaling pathways is additionally supported by results showing that both mouse and human SEF mediates a reduction in AKT signaling [120,121]. Overexpression of SEF in the lens of mice resulted in impaired lens and eye development and increased apoptosis due to inhibition of FGFR signaling during lens morphogenesis [122]. The intracellular, extracellular and the transmembrane domain of SEF seem to be important for SEF-mediated negative regulation of FGF signaling [123].

Another protein that might modulate FGFR signaling is FGFR5 (FGFRL1). FGFR5 is the fifth member of the FGFR family but lacks tyrosine kinase activity and thus is not an active receptor. Instead, FGFR5 has been predicted to negatively regulate FGF signaling by competing with other FGFRs for ligand binding or by forming heterodimers with other members of the FGFR family and thereby preventing trans-autophosphorylation [124]. However, experimental results showed that FGFR5 does not function as a decoy receptor, but rather promotes intracellular signaling as it increases activation of MAPK signaling due to association with the phosphatase SHP1 [125]. Flotillin-1, a multifunctional protein also involved in endocytosis, has been shown to compete with FGFR for binding to FRS2 and thereby interferes with signaling [126]. A flotillin-1 knockdown resulted in increased tyrosine phosphorylation of FRS2, as well as inhibition of ERK activity. 

Grb2 has also been suggested to play a role as a regulator of FGFR2 signaling [127]. It was reported that dimeric Grb2 can bind directly to unliganded FGFR2, preventing receptor phosphorylation. Upon stimulation, the activated FGFR2 phosphorylates Grb2, leading to Grb2 dissociation and the full activation of FGFR2 downstream signaling.

The adaptor protein Grb14 has been implicated in the regulation of PLCγ signaling downstream of FGFRs [128]. Upon receptor activation, PLCγ is directly recruited to phosphorylated tyrosine 766 in FGFR1 [129]. Similarly, Grb14 was found to bind to the same phosphorylated tyrosine in FGFR1 (Y766) and is thought to compete with PLCγ for binding [130]. In addition, it was suggested that the binding of Grb14 to phosphorylated Y766 in FGFR1 led to conformational changes in Grb14 that unmasked a PLCγ binding motif in Grb14 [131]. Once the binding motif is revealed, PLCγ is trapped away from the active FGFR and cannot be phosphorylated. Thus, Grb14 seems to have a dual way of inhibiting FGFR-induced PLCγ signaling. 

Clearly, multiple proteins can function at various steps as negative regulators in the FGFR signaling pathways to turn off FGFR signaling. In addition, some of these regulatory proteins might have numerous roles.

## 5. Negative Feedback Phosphorylations

In addition to several negative feedback mechanisms governing FGFR signaling such as FGF-induced CBL recruitment, MKP3 production and others discussed above, more direct negative feedback mechanism involving phosphorylation events also exists to prevent excess FGFR signaling (Figure 1). Nearly all components of the MAPK signaling pathways are regulated through such negative feedback phosphorylations by downstream kinases [97]. For example, upon activation, FGFRs phosphorylate FRS2α on tyrosine residues leading to recruitment of Grb2 and activation of MAPK signaling and ERK1/2 activation. Active ERK1/2 can then phosphorylate eight threonine residues in FRS2 [132]. Mutation of the FRS2α threonine phosphorylation sites led to constitutive tyrosine phosphorylation of FRS2α in unstimulated cells and enhanced FRS2α tyrosine phosphorylation in FGF-stimulated cells [132]. Similarly, a specific serine residue (S777) in the C-terminal region of FGFR1 was shown to be directly phosphorylated by active ERK1 and ERK2 upon FGF1 stimulation [133]. Mutating S777 to alanine led to prolonged FGFR1 tyrosine phosphorylation indicating a direct negative feedback mechanism attenuating FGFR1 signaling [133]. A negative feedback loop mediated by ERK1/2 pathway was also identified for FGFR2 [134]. It was shown that substituting serine 780 in FGFR2 (corresponding to S777 in FGFR1) with alanine resulted in increased FGFR2 phosphorylation and signaling. Hence, ERK1/2-mediated phosphorylation of S780 in FGFR2, similarly to S777 in FGFR1, acts as a negative feedback loop to prevent excess signaling. It is not known how phosphorylation of S777 (FGFR1) and S780 (FGFR2) regulate receptor activity. It is also worth mentioning that, a serine close to S780 in FGFR2 (S782 or S779 dependent on the numbering) is phosphorylated by PKCε, providing a docking site for the adaptor protein 14-3-3 and leading to sustained ERK activation [135]. It is not known how these negative and positive feedback phosphorylation events in FGFR2 cooperate to regulate FGFR2 signaling. Interestingly, the serine/threonine kinase RSK2, which is activated downstream of ERK1/2, can bind directly to FGFR1 and phosphorylate serine 789 (S789) in FGFR1 [42]. Phosphorylation of S789 in FGFR1 is important for proper FGFR1 endocytosis and ubiquitination [42].

Lack of such negative feedback loops could give tumor cells a growth advantage and indeed, mutants of FGFR2 lacking S780 have been identified in human cancers [134,136,137] indicating a crucial role of negative feedback phosphorylation loops in FGFR signaling. 

## 6. Summary

Clearly, many mechanisms exist to control FGFR signaling. The tight regulation of signaling is necessary to maintain body homeostasis. These mechanisms can occur simultaneously or sequentially, and in different or, sometimes, the same steps of the signaling pathways. They might operate differently depending on cell type, the ligand or the receptor type. It is clear that our cells have evolved a whole menu of mechanisms reinforcing each other to control FGFR signaling. In case one of the mechanisms is not operational, other mechanisms and proteins involved in the regulation of cell signaling can take over and the cell might still be able to prevent excessive FGFR signaling. The development of all of these different mechanisms indicates the importance of keeping FGFR signaling under control. As mentioned in the introduction, several human conditions are caused by excessive FGFR signaling, among them dwarfism and cancer [15,16,17,18]. Clearly, the magnitude, duration, and location of FGFR signaling must be strictly controlled in order to induce the correct biological response and prevent development of diseases. 

## Figures and Tables

**Figure 1 cells-10-01342-f001:**
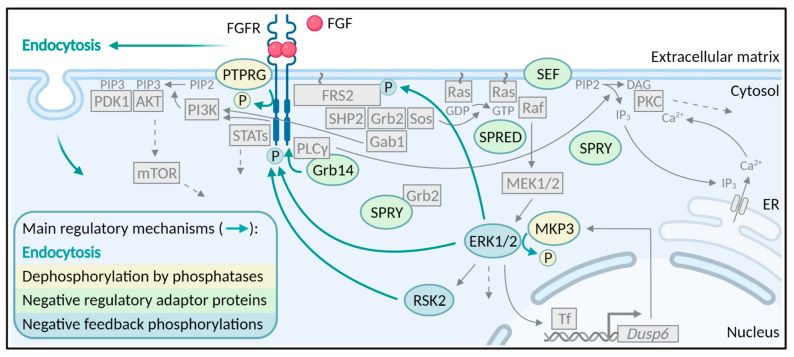
Turning off FGFR signaling. Endocytosis, dephosphorylation by phosphatases, inhibitory adaptor proteins competing for target and negative feedback phosphorylation loops are all mechanisms that act to turn off FGFR signaling. The four main FGFR signaling pathways, Ras-MAPK, PI3K-AKT, PLCγ-PKC and STAT pathway are indicated in gray. Green arrows indicate a negative regulatory mechanism. Tf, transcription factor, ER, endoplasmic reticulum. Created with BioRender.com, accessed on 1 May 2021.

**Figure 2 cells-10-01342-f002:**
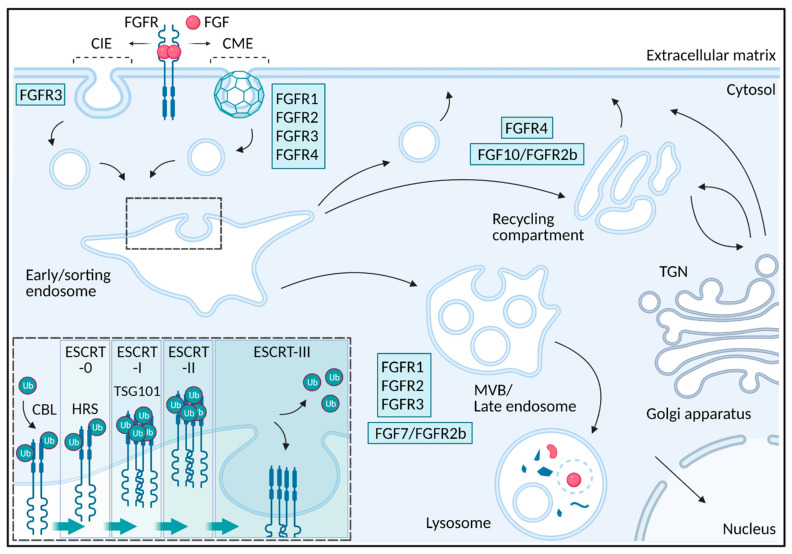
Endocytosis of FGFs/FGFRs. Once activated by ligand-binding, FGFRs can be internalized via clathrin-mediated endocytosis (CME) and clathrin-independent endocytosis (CIE). While FGFR1, 2 and 4 are mainly internalized via CME, FGFR3 seems to be internalized partly by CME and partly by CIE mechanisms. Once internalized, the ligand-receptor complexes are localized to early/sorting endosomes from which they can be sorted to recycling either directly or via the endocytic recycling compartment or to degradation in lysosomes via multivesicular bodies (MVB) and late endosomes. FGFRs destined for degradation in lysosomes are tagged by the attachment of ubiquitin (Ub). Ubiquitination of FGFRs seems to be dependent on the E3 ubiquitin ligase CBL. Ubiquitinated receptors are then recognized by the ESCRT complexes (ESCRT-0-III) and sorted into intraluminal vesicles originating at the endosomal membrane. Ub is removed before internalization. At least two components of the ESCRT machinery have proven important for FGFR sorting into the degradative pathway, namely HRS of ESCRT-0 and TSG101 of ESCRT-I. The decision to degrade or not, depends on the receptor type as well as the bound ligand. FGFR4 seems to be mainly recycled while FGFR1, 2 and 3 are sorted more efficiently to lysosomal degradation. FGFR2b bound to FGF7 is sorted for degradation while FGFR2b bound to FGF10 is recycled. In both cases the decision to degrade or not, seems to depend on the levels of receptor ubiquitination. The recycling of FGFR4 might also occur via TGN (trans Golgi network). Moreover, both FGFs and FGFRs can translocate to the nucleus. Created with BioRender.com, accessed on 1 May 2021.

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
