# Peer review of "Negative Regulation of FGFR (Fibroblast Growth Factor Receptor) Signaling"

_cells, 2021, doi:10.3390/cells10061342_

Round 1
Reviewer 1 Report
This is a well-written and very useful review on an aspect of FGF biology, which has not been extensively covered in previous reviews. It is important for the FGF field, but also for other RTKs. The authors are major experts in this area. I have only very few comments/suggestions:
The authors mention at several places that the cell type is highly relevant for the fate of the receptor after internalization or for other negative regulatory mechanisms. Therefore, I encourage the authors to mention (wherever possible) in which cell type a study was performed. In particular, it is important if a mechanism was identified for the endogenous receptor in its natural context or if it was shown in heterologous expression systems with overexpressed receptor, which may not normally be expressed in this cell type. This could strongly affect the result. This could also be summarized in a table.
I suggest to replace “KGFR” by FGFR2b – this is the official name.
Finally, a few typos should be corrected, e.g. pade 11 , line 484 (signaling instead of singling).
Author Response
“The authors mention at several places that the cell type is highly relevant for the fate of the receptor after internalization or for other negative regulatory mechanisms. Therefore, I encourage the authors to mention (wherever possible) in which cell type a study was performed. In particular, it is important if a mechanism was identified for the endogenous receptor in its natural context or if it was shown in heterologous expression systems with overexpressed receptor, which may not normally be expressed in this cell type. This could strongly affect the result. This could also be summarized in a table.”
Reply: We appreciate the suggestion to include which cell lines were used wherever possible and also to state whether or not the receptor was naturally occurring in that cell line or rather ectopically expressed. We agree that this is desirable for the reader to know, especially in the chapter about endocytic processes as it is a well-known fact that studies regarding endocytosis should be interpreted with care and that receptor levels might interfere with the results. We fear however, that the article will be too packed with details and not easily readable if the information is included in the text.
“I suggest to replace “KGFR” by FGFR2b – this is the official name.”
Reply: We have now replaced KGFR with FGFR2b in the text and in the figure. We have also included a small paragraph in the introduction to explain the FGFR b and c forms (line 39-44).
“Finally, a few typos should be corrected, e.g. pade 11 , line 484 (signaling instead of singling)”
Reply: Thank you for noting these. We have now corrected the typo in page 11, line 484 (old version) and also gone carefully through the manuscript to eliminate others.
Reviewer 2 Report
The authors focus on the not abundantly studied, yet important field of negative regulations of FGFR signaling. Within their review article, they present four major mechanisms of such regulations - intracellular sorting/endocytosis, action of phosphatases, negative regulatory proteins, and feedback phosphorylations. The manuscript is written with attention to detail, and I do appreciate that the authors do not write exclusively about their line of research. While I think the study has a potential to be published in the Journal, there are several things that need to be addressed.
First, the manuscript would benefit from information obtained in animal models or human patients in which the negative regulatory mechanism was not functional, due to knock-out or loss-off-function mutation for example. Especially the authors should focus on the resulting phenotype that could be explained, or even was shown to be caused by deregulated FGFR signaling. This is because cell line experiments often present a phenotype, that however does not translate into a functional relevance in the living organism, due to compensatory mechanisms for example. It would be interesting to read about the developmental/homeostatic processes for which the negative FGFR regulation is important within this review article.
Second, it would be ineteresting to learn if different subcellular localization of the FGFR complexes alters the possibilities of negative feedback mechanisms. FGFRs have been shown to localize to different cellular compartments, either through their physiological function (tight junctions, primary cilia...), or due to features gained via genetic mutation (persistent presence in the endoplasmic reticulum, or saturation at the centrosome due to cell fusion for example). While I am no expert here, I can imagine that different cellular compartments would allow for distinct mechanismis of receptor activation, signaling and silencing. The authors should reflect these possibilities in their manuscript, sticking only to the negative regulatory mechanisms of course.
Third, the FGFR5, or FGFRL1 has been speculated to play a negative regulatory function in the FGFR pathways. The authors should reflect that in their manuscript.
Additional minor point:
The manuscript text is sometimes very rich in mechanistic information. This is understandable, since the article is focused on signaling mechanisms; yet sometimes it becomes quite difficult to follow. The authors might consider adding figures that would help to understand all mentioned regulatory mechanisms more clearly.
Alltogether, the manuscript follows an interesting topic of FGFR biology and I think it should be published, once the above mentioned point are adressed.
Author Response
“First, the manuscript would benefit from information obtained in animal models or human patients in which the negative regulatory mechanism was not functional, due to knock-out or loss-off-function mutation for example. Especially the authors should focus on the resulting phenotype that could be explained, or even was shown to be caused by deregulated FGFR signaling. This is because cell line experiments often present a phenotype, that however does not translate into a functional relevance in the living organism, due to compensatory mechanisms for example. It would be interesting to read about the developmental/homeostatic processes for which the negative FGFR regulation is important within this review article.”
Reply: We agree with the reviewer that it is very interesting and important to know if these negative regulative mechanisms have been shown to be significant in actual processes in vivo (development etc). Although this is not our field of expertise, we have searched through the literature and found examples of experiments performed in animal models showing defects due to malfunction of the negative regulation of FGFR signaling. These are now included in the manuscript for endocytosis (line 325-346), phosphatases (p. 388-394), and negative regulatory proteins (line 423-424, 445-449, 472-474). For negative feedback mechanisms we were not able to find any except for the possible relevance in cancer mentioned already (line 534-537).
“Second, it would be interesting to learn if different subcellular localization of the FGFR complexes alters the possibilities of negative feedback mechanisms. FGFRs have been shown to localize to different cellular compartments, either through their physiological function (tight junctions, primary cilia...), or due to features gained via genetic mutation (persistent presence in the endoplasmic reticulum, or saturation at the centrosome due to cell fusion for example). While I am no expert here, I can imagine that different cellular compartments would allow for distinct mechanismis of receptor activation, signaling and silencing. The authors should reflect these possibilities in their manuscript, sticking only to the negative regulatory mechanisms of course.”
Reply: This is also an excellent point raised by the reviewer. It would be very interesting to know if the different mechanisms are reserved for specific locations within the cell. To our knowledge, we found at least one example of this, concerning FGFR phosphatases. This is now included in the manuscript (line 362-364).
“Third, the FGFR5, or FGFRL1 has been speculated to play a negative regulatory function in the FGFR pathways. The authors should reflect that in their manuscript.”
Reply: The potential role of FGFR5 in negative regulation of FGFR signaling is now included in the manuscript (477-484).
“The manuscript text is sometimes very rich in mechanistic information. This is understandable, since the article is focused on signaling mechanisms; yet sometimes it becomes quite difficult to follow. The authors might consider adding figures that would help to understand all mentioned regulatory mechanisms more clearly.”
Reply: We agree that additional figures would be a way to help the reader to follow all aspects of the different mechanisms, especially in the field of negative regulatory proteins. However, as many of these mechanisms are not well defined and the actual acting point of the mechanisms are not well established (i. e. SPRED in MAPK pathway, SEF, Sprouty) we found it somewhat difficult. We also feel that highlighting these mechanisms in a figure, without a stronger consensus in the field might be a too strong statement compared to mentioning them as possible mechanisms in the text. We have, however, slightly changed Figure 1 (increased arrow size etc) to underline the different mechanisms and proteins involved.